# Optimizing Parameters in the Common Land Model by Using Gravity Recovery and Climate Experiment Satellite Observations

Yuan Su and Shupeng Zhang *

Southern Marine Science and Engineering Guangdong Laboratory (Zhuhai), School of Atmospheric Sciences,
Sun Yat-sen University, Zhuhai 519082, China; suyuan6@mail2.sysu.edu.cn
* Correspondence: zhangshp8@mail.sysu.edu.cn

**Abstract:** Terrestrial water storage (TWS) is pivotal in understanding environmental dynamics, climate change, and human impacts. Despite the utility of land surface models, uncertainties persist in their parameterization schemes. This study employs GRACE (Gravity Recovery and Climate Experiment) satellite data to optimize the runoff parameterization scheme within the Common Land Model by a data assimilation and parameter optimization method. The optimization algorithm sets an adjustment factor that varies with time and space for runoff simulation and updates it along with the running of the land surface model. The evaluation reveals that there are improved correlation coefficients and reduced root mean square errors compared to GRACE observations. Independent assessments by using in situ river discharge observations demonstrate enhanced model performance, particularly in mountainous regions such as western North America. This study underscores the efficacy of integrating GRACE data to improve land surface model parameterization, offering more accurate predictions of TWS changes.

**Keywords:** land surface model; GRACE satellite; runoff parameterization scheme; parameter optimization





## 1. Introduction

Terrestrial water storage (TWS) is a key variable in the earth system [1]. It has two-way feedback with climate change [1,2], and variations in TWS strongly impact carbon sinks through the interactions between water and carbon cycles [3].

Land surface models can simulate the state of TWS and its variability due to changes in precipitation, evapotranspiration, soil water movement, and runoff [4,5]. Among these fluxes, runoff reflects the comprehensive effects of climate, topography, and land surface characteristics at different spatial and temporal scales, exhibiting strong heterogeneity [6,7]. Runoff fluxes play an important role in hydrological cycles and, thus, in the earth system. On the one hand, runoff simulation is influenced by TWS and the groundwater table [8]. On the other hand, improvements in runoff simulation can affect other components of the water cycle through water balance [9,10].

In land surface models, there are primarily two approaches for simulating runoff: parameterization schemes and an explicit incorporation of lateral flow dynamics [11]. Explicitly considering lateral flow enhances the accuracy of water cycle simulations [9,10]. However, this method necessitates data and parameterization schemes at the hillslope scale, which are currently limited [6]. As a result, simplistic parameterization schemes remain prevalent in most land surface simulations, particularly in global or climate models. However, these oversimplified schemes often introduce significant uncertainties in runoff estimation.

Combining observational data with land surface models holds promise for enhancing the estimates of TWS, runoff, and simulations of water cycle processes [12]. Most land surface models lack descriptions of deeper groundwater and do not account for changes in lake and river water storage, necessitating calibration or validation with observational data [13].

Furthermore, Gravity Recovery and Climate Experiment (GRACE) satellite datasets are extensively utilized for monitoring TWS changes [14–18], assessing and evaluating land surface models [19–22], and performing data assimilation and parameter calibration [23–29]. GRACE satellite observations capture TWS changes and can effectively constrain water cycle dynamics within land surface models [5]. Conversely, simulation outputs from land surface models enable the disentanglement of TWS components observed by GRACE. Additionally, discrepancies between TWS components simulated by models and those observed by GRACE can help identify abnormal changes in large-scale groundwater storage globally, facilitating assessments of human activities' impacts such as irrigation, pumping, and reservoir operations on TWS [14,20].

In this study, GRACE satellite observations were integrated into the Common Land Model (CoLM) [30] to enhance its water cycle simulations. A data assimilation scheme employing an optimal estimation of model errors was developed. To maintain mass conservation, scaling factors were introduced and updated in the runoff parameterization scheme during the assimilation step, rather than updating the states of the TWS components.

Section 2 outlines the data and methods employed in this study. The results are presented in Section 3 and subsequently discussed. Finally, Section 4 provides the conclusions drawn from the study.

## 2. Model, Data and Methods

Data assimilation algorithms are methods that combine model predictions with observation data. In this section, the framework of assimilating GRACE observations for parameter optimization in the Common Land Model is introduced.

### 2.1. Land Surface Model

CoLM [30] simulates energy, water, biogeochemical, and other land processes. In CoLM, terrestrial water storage (TWS) is calculated as the sum of water storage in canopy, snow, land ice, surface water, soil moisture, and water in an aquifer. The water balance equation within each grid cell is written as follows:

$$\frac{\Delta W}{\Delta t} = P - ET - R \tag{1}$$

where $\Delta W$ represents the change in TWS within a grid cell; $\Delta t$ denotes a certain period; $P$ and $ET$ indicates the precipitation and evapotranspiration rate over the same period, respectively; and $R$ represents the total runoff rate. Precipitation is obtained from atmospheric forcing data, while evapotranspiration and runoff are simulated by the model.

Parameterization schemes are used for the simulations of runoff in this study [31]. The total runoff consists of surface runoff and subsurface runoff. The parameterization scheme for surface runoff considers factors such as groundwater table depth, precipitation, and saturated hydraulic conductivity in soil. Within a model grid cell, all of the surface water in the saturated area contributes to runoff, while in the unsaturated area, only excess water beyond infiltration contributes to runoff. The fraction of saturated area within the model grid cell is calculated as follows [31]:

$$f_{\text{sat}} = f_{\text{wt}} \cdot e^{-0.5 \cdot f_{\text{decay}} \cdot z_{\text{wt}}} \tag{2}$$

where $z_{\text{wt}}$ represents the groundwater table depth (m), $f_{\text{wt}}$ represents the fraction of area in the grid cell with shallow groundwater depth, and $f_{\text{decay}}$ is a decay factor. $f_{\text{wt}} = 0.38$ and $f_{\text{decay}} = 0.5 \, \text{m}^{-1}$ are taken as constants in the scheme.

In the unsaturated area, the maximum infiltration capacity considers the state and properties of the top three layers of soil and is calculated as follows [31]:

$$q_{\text{in,max}} = \min_{i=1,2,3} \left\{ \left( 10^{-6.0 \cdot f_{\text{ice},i}} \right) \cdot K_{\text{sat},i} \right\} \tag{3}$$

where $f_{\text{ice},i}$ represents the volume percentage of ice in the soil pores for the *i*th layer of soil, and $K_{\text{sat},i}$ represents the saturated hydraulic conductivity of the *i*th layer of soil.

The total surface runoff is then estimated as follows [31]:

$$r_{\text{surface}} = f_{\text{sat}} \cdot G_{\text{wat}} + (1 - f_{\text{sat}}) \cdot \max(G_{\text{wat}} - q_{\text{in,max}}, 0) \tag{4}$$

where $G_{\text{wat}}$ represents the amount of liquid water reaching the surface, including rain through the canopy, water dripping from the canopy, and snowmelt; $f_{\text{sat}}$ represents the fraction of the grid cell area that is saturated by (2); and $q_{\text{in,max}}$ represents the maximum infiltration capacity calculated by (3).

The subsurface runoff is estimated by [31]

$$r_{\text{subsurface}} = q_{\text{drai,max}} \cdot \exp(-f_{\text{drai}} \cdot z_{\text{wt}}) \tag{5}$$

where $q_{\text{drai,max}}$ is the maximum value of drainage, and $f_{\text{drai}} = 2.5 \, \text{m}^{-1}$ is the decay factor. When soil ice is present, the resistance of ice is considered. $q_{\text{drai,max}}$ takes a globally uniform value of $5.5 \times 10^{-3}$ mm/s in the scheme.

Globally, runoff depends on factors such as topography, soil properties, and land cover types, which are neglected in the above simplistic parameterization schemes. Although parameters in these schemes can achieve a certain level of simulation accuracy after parameter calibration, large uncertainties still exist due to the neglect of key factors. This study optimizes the runoff parameterization scheme further by using GRACE observation data.

CoLM requires meteorological forcing data including variables of wind speed, air temperature, humidity, air pressure, precipitation, shortwave radiation, and longwave radiation. In this study, these atmospheric forcing data are obtained from the CRU JRA dataset v2.4, which is a gridded land surface blend of Climatic Research Unit and Japanese reanalysis data [32]. The CRU JRA dataset provides 6-hourly 0.5-degree gridded data covering the global region except for Antarctica, with a time series from 1901 to 2020.

The spatial resolution of the CoLM simulation is 2 degrees, and the forcing data are upscaled from 0.5 degree to 2 degrees using an area weighted averaging method. The temporal resolution is half-hourly. The resolution of 2 degrees is used in this study because GRACE products have a native resolution of 3 degrees [33] and allow for a spatial resolution of typically 300 km [34].

To ensure the model reaches a near-equilibrium state of water, a spin-up with approximately 100 years was carried out before assimilating GRACE data. The near-equilibrium state was determined based on the criterion that the ten-year moving average of the change in $\Delta W$ was less than 2% of the annual precipitation. It was observed that over 90% of the land grid cells with annual precipitation exceeding 250 mm attained a state of near-equilibrium after the spin-up period.

The spin-up period was from 1901 to 2001, and data assimilation started in 2002 when GRACE observation was available. Monthly averaged values of TWS by CoLM were calculated on each grid to make them consistent with the GRACE observations.

To evaluate the data assimilation results through independent observations, CoLM was coupled with CaMa-Flood [35] to simulate discharge in rivers. The simulated discharge was compared with in situ observation data on a global scale from the Global Runoff Database Center (GRDC).

### 2.2. Observations

The monthly averaged TWS data in the JPL GRACE and GRACE-FO MASCON RL06Mv2 CRI datasets was used [33,36] in this study. The JPL Mascon data do not use empirical filtering for noise reduction but include additional geophysical information beyond the GRACE data. Moreover, a coastal resolution improvement filter has been developed to address coastline issues. The JPL-RL06M product is represented on 0.5-degree grid, while the effective resolution is 3 degrees. The datasets cover a time period of 2002 to 2022, but only observations from 2002 to 2020 were used in this study.

The GRACE-derived difference of TWS in two consecutive months was taken as an observation in the data assimilation framework. The difference between the two observations is written as

$$\Delta W^{\mathrm{o}} = W_1^{\mathrm{o}} - W_0^{\mathrm{o}} \tag{6}$$

where $W_0^{\mathrm{o}}$ and $W_1^{\mathrm{o}}$ are observed TWS by GRACE in months $T_0$ and $T_1$, with uncertainties of $\sigma_0^{\mathrm{o}}$ and $\sigma_1^{\mathrm{o}}$, respectively. Assuming the two observations are independent, the variance of $\Delta W^{\mathrm{o}}$ is given by $(\sigma^{\mathrm{o}})^2 = (\sigma_1^{\mathrm{o}})^2 + (\sigma_0^{\mathrm{o}})^2$. Using $\Delta W$ between two consecutive months removes the long-term trend of TWS changes, which is beneficial for optimizing runoff schemes.

The GRDC dataset (https://grdc.bafg.de/GRDC/EN/Home/homepage_node.html, accessed on 3 April 2024) was used in this study to evaluate the model's simulated discharge; while the dataset contains river discharge from over 10,000 stations globally, only observations from those with an upstream area larger than 10,000 km$^2$ were selected since the model is run at coarse resolution. The evaluation time period was from 2006 to 2015, which is after a 4-year data assimilation started in 2002, and monthly averaged values were used.

*2.3. Data Assimilation Method*

The forecast value of $\Delta W$ by CoLM is denoted as $\Delta W^{\mathrm{f}}$. With an unbiased estimator for a scalar state variable combined with a single measurement [37], the analysis value of $\Delta W$, denoted as $\Delta W^{\mathrm{a}}$, can be calculated from $\Delta W^{\mathrm{o}}$ and $\Delta W^{\mathrm{f}}$ by

$$\Delta W^{\mathrm{a}} = \frac{\Delta W^{\mathrm{f}}(\sigma^{\mathrm{o}})^2 + \Delta W^{\mathrm{o}}\left(\sigma^{\mathrm{f}}\right)^2}{\left(\sigma^{\mathrm{f}}\right)^2 + (\sigma^{\mathrm{o}})^2} \tag{7}$$

where $\left(\sigma^{\mathrm{f}}\right)^2$ denotes the variance of $\Delta W^{\mathrm{f}}$.

In order to maintain the mass conservation of water, a factor $\alpha_{\mathrm{slp}}$ to adjust the simulated runoff was introduced instead of updating the TWS state in CoLM, satisfying

$$\frac{\Delta W^{\mathrm{a}}}{\Delta t} = P - ET - \alpha_{\mathrm{slp}} \cdot R \tag{8}$$

where the subscript "slp" in $\alpha_{\mathrm{slp}}$ indicates that the error in the runoff is mainly from the model's neglect of the influence of topography. Considering that runoff exhibits significant seasonal variations while satellite observation data are monthly values, $\alpha_{\mathrm{slp}}$ is further divided into 12 values, one for each month.

The online updating scheme for the adjustment factor $\alpha_{\mathrm{slp}}$ is as follows: (1) initialize the value of $\alpha_{\mathrm{slp}}$ to 1; (2) in each two consecutive months $m$ and $m + 1$ with satellite observations, run the model based on the current values of $\alpha_{\mathrm{slp},m}$ and $\alpha_{\mathrm{slp},m+1}$ to forecast $\Delta W^{\mathrm{f}}$, and then combine $\Delta W^{\mathrm{f}}$ with the observed $\Delta W^{\mathrm{o}}$ to update the adjustment factors $\alpha_{\mathrm{slp},m}$ and $\alpha_{\mathrm{slp},m+1}$; (3) continue running the model to the next observation months with satellite observations, returning to step 2. The workflow is shown in Figure 1.

The variance of the forecast $\left(\sigma^{\mathrm{f}}\right)^2$ is estimated with the maximum likelihood method [38]. Assuming $\Delta W^{\mathrm{f}} - \Delta W^{\mathrm{o}}$ is a random variable with mean 0 and variance $\left(\sigma^{\mathrm{f}}\right)^2 + (\sigma^{\mathrm{o}})^2$ [39], the -2log-likelihood function for $\left(\sigma^{\mathrm{f}}\right)^2$ is given by

$$-2L\left(\left(\sigma^{\mathrm{f}}\right)^2\right) = \ln\left[\left(\sigma^{\mathrm{f}}\right)^2 + (\sigma^{\mathrm{o}})^2\right] + \frac{(\Delta W^{\mathrm{f}} - \Delta W^{\mathrm{o}})^2}{\left(\sigma^{\mathrm{f}}\right)^2 + (\sigma^{\mathrm{o}})^2} \tag{9}$$

When $-2L$ attains its minimum value, $\left(\sigma^{\mathrm{f}}\right)^2$ satisfies the equation

$$\frac{1}{(\sigma^{\mathrm{f}})^2 + (\sigma^{\mathrm{o}})^2} - \frac{(\Delta W^{\mathrm{f}} - \Delta W^{\mathrm{o}})^2}{\left[(\sigma^{\mathrm{f}})^2 + (\sigma^{\mathrm{o}})^2\right]^2} = 0 \qquad (10)$$

It can be solved for $\left(\sigma^{\mathrm{f}}\right)^2$ from (10):

$$\left(\sigma^{\mathrm{f}}\right)^2 = \left(\Delta W^{\mathrm{f}} - \Delta W^{\mathrm{o}}\right)^2 - (\sigma^{\mathrm{o}})^2 \qquad (11)$$

The estimated $\left(\sigma^{\mathrm{f}}\right)^2$ from (11) is then used in (7) for the calculation of the analysis value. It is noteworthy that when the observation error $\sigma^{\mathrm{o}}$ is greater than $|\Delta W^{\mathrm{f}} - \Delta W^{\mathrm{o}}|$, the solution (11) is meaningless. In this case, the observation is considered invalid and discarded.

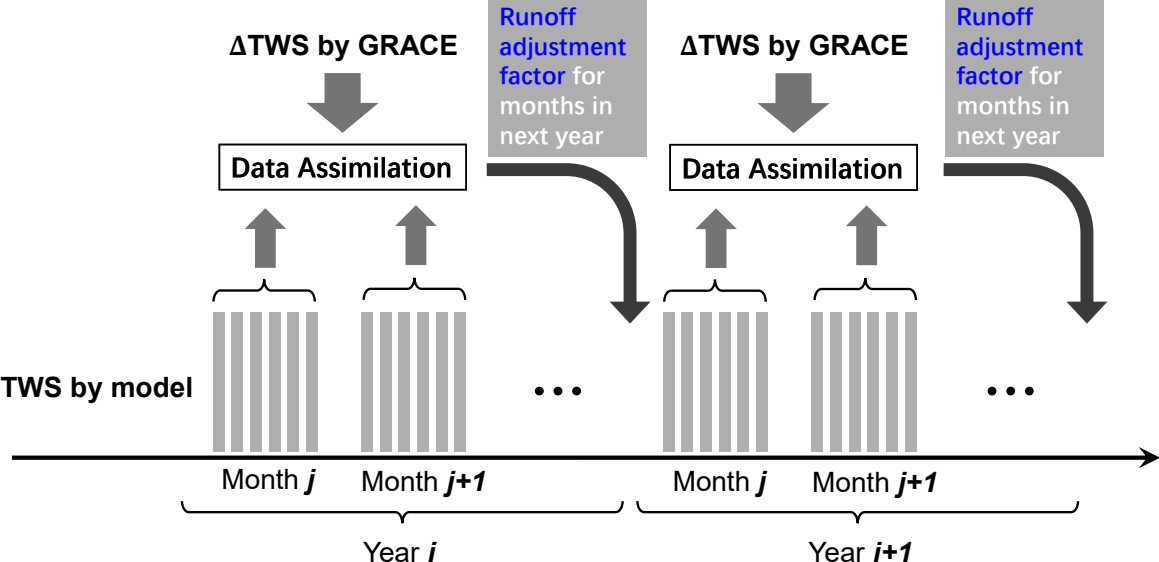

**Figure 1.** Workflow of data assimilation algorithm.

*2.4. Evaluations*

The following three statistical metrics are used to evaluate the results of parameter optimization. The Pearson correlation coefficient is calculated by

$$r = \frac{\sum_{i=1}^{n}(y_i - \bar{y})(o_i - \bar{o})}{\sqrt{\sum_{i=1}^{n}(y_i - \bar{y})^2}\sqrt{\sum_{i=1}^{n}(o_i - \bar{o})^2}} \qquad (12)$$

where $o_i$ represents the time series of the observed $\Delta W$ from GRACE observations; $y_i$ represents the time series of modeled $\Delta W$; $n$ is the number of data in the time series; $\bar{o}$ and $\bar{y}$ are the means of $o_i$ and $y_i$, respectively. The correlation coefficient $r$ ranges from $-1$ to 1, where a larger value indicates a higher correlation between the simulations and observations. After adjusting the parameters in the model, if the correlation coefficient improves, then it is considered as parameter optimization.

The Root Mean Square Error (RMSE) is calculated by

$$\mathrm{RMSE} = \sqrt{\frac{1}{n}\sum_{i=1}^{n}(y_i - o_i)^2} \qquad (13)$$

where $o_i$ represents the observed values of $\Delta W$ from GRACE observations; $y_i$ represents the modeled values of $\Delta W$; $n$ is the number of data in the time series. A smaller value of RMSE indicates a higher degree of closeness between the simulations and observations.

After adjusting the parameters in the model, if the RMSE decreases, then it is considered as parameter optimization.

The Nash–Sutcliffe efficiency coefficient is used for the comparisons and is defined as

$$\text{NSE} = 1 - \frac{\sum_{i=1}^{n}(y_i - o_i)^2}{\sum_{i=1}^{n}(o_i - \bar{o})^2} \tag{14}$$

where $o_i$ represents the observed discharge from the GRDC dataset; $y_i$ represents the modeled discharge by using coupled CoLM and CaMa-Flood; $n$ is the number of data in the time series. A value of NSE closer to 1 indicates a higher degree of closeness between the simulations and observations. In this study, observed discharge data spanning from 2006 to 2015 were utilized, with monthly averaged values derived for analysis.

### 3. Results and Discussions

This section presents the results of parameter optimization using GRACE satellite data and gives a preliminary analysis of the results.

Since $\alpha_{\text{slp}}$ functions as a multiplication factor, the investigation focuses on the time-averaged logarithm (with base e):

$$\log \bar{\alpha}_{\text{slp}} = \frac{1}{n} \sum_{i=1}^{n} \log(\alpha_{\text{slp},i}) \tag{15}$$

where $n$ is the number of data in the time series. A positive value of $\log \alpha_{\text{slp}}$ indicates an underestimation of runoff in the forecast, while a negative value indicates an overestimation. Figure 2 shows global $\log \bar{\alpha}_{\text{slp}}$ values averaged from 2002 to 2020. It can be seen that $\log \bar{\alpha}_{\text{slp}}$ varies both positively and negatively globally. Positive values are concentrated in regions with low precipitation, such as western North America, northern Africa, northwestern China, Greenland, etc., while negative values are concentrated near the equator and in high-latitude regions such as northern North America and northern Eurasia. Regions with significant variations, such as around the Tibetan Plateau and western North America, are typical areas with significant terrain fluctuations.

Figures 3 and 4 depict the annual mean runoff without data assimilation and the differences observed upon assimilating GRACE data. Through GRACE data assimilation, an increase in runoff is evident across most areas between 30° S and 30° N, as well as regions north of 60° N. Conversely, notable decreases in runoff are observed in western North America and eastern Europe. An increase in runoff generally indicates an overestimated TWS change in wet seasons or underestimated TWS change in dry seasons by the model compared to GRACE observations. For example, the largest increase in Islands near the equatorial Pacific is due to an overestimated TWS change in wet seasons (not shown here). By contrast, a decrease in runoff is usually the result of an underestimated TWS change in wet seasons or overestimated TWS change in dry seasons. As an example, runoff in western North America is decreased because of an underestimated TWS change in wet seasons.

Figure 5 illustrates the correlation coefficients defined in (12) without parameter optimization. The correlation coefficients between the simulated TWS changes and observations are greater than 0.5 for more than half of the global land grid points (depicted in red and yellow). Grid points with lower correlation coefficients are mostly located in regions with a low annual average precipitation (such as the Sahara Desert region, northwestern China) or regions with a high annual average precipitation (such as the equatorial region in the Pacific and Amazon River basin). In these areas (depicted in blue), the lower correlation coefficients may be attributed to the smaller variations in TWS or less pronounced seasonal changes.

Figure 6 illustrates the difference between correlation coefficients resulting from assimilating GRACE data into CoLM and simulations without data assimilation. With parameter optimization, the global distribution of correlation coefficients between simulated TWS changes and observations remains largely similar to that without optimization. Approx-

imately 62% of grid points globally exhibit an enhancement in correlation coefficients, while 36% witness a decline compared to simulations without optimization. Significant enhancements in correlation coefficients are notably observed in regions near 60° N, eastern China, and upstream areas of the Amazon River basin. Conversely, prominent declines in correlation coefficients are evident in central parts of the Amazon River basin, southern regions of the Sahara Desert, and certain areas of the Tibetan Plateau.

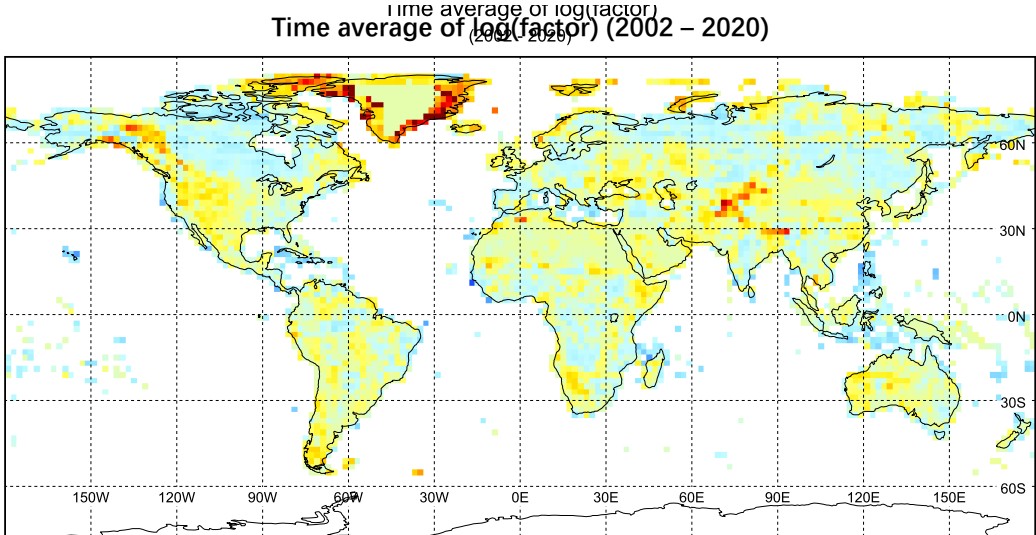

**Figure 2.** Time-averaged logarithm of adjustment factor to runoff in CoLM.

**Figure 3.** Annual mean runoff simulated by CoLM without data assimilation.

Difference in Annual Mean Runoff (With DA minus Without DA)

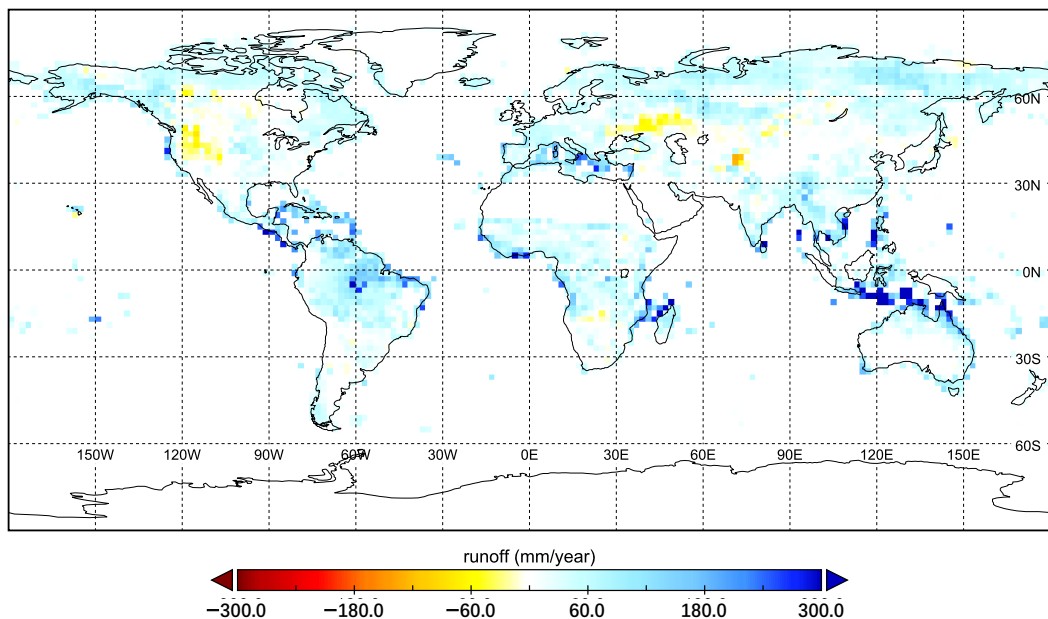

**Figure 4.** Difference in annual mean runoff between simulations with and without assimilating GRACE data into CoLM.

Correlation Coefficients - Without Data Assimilation

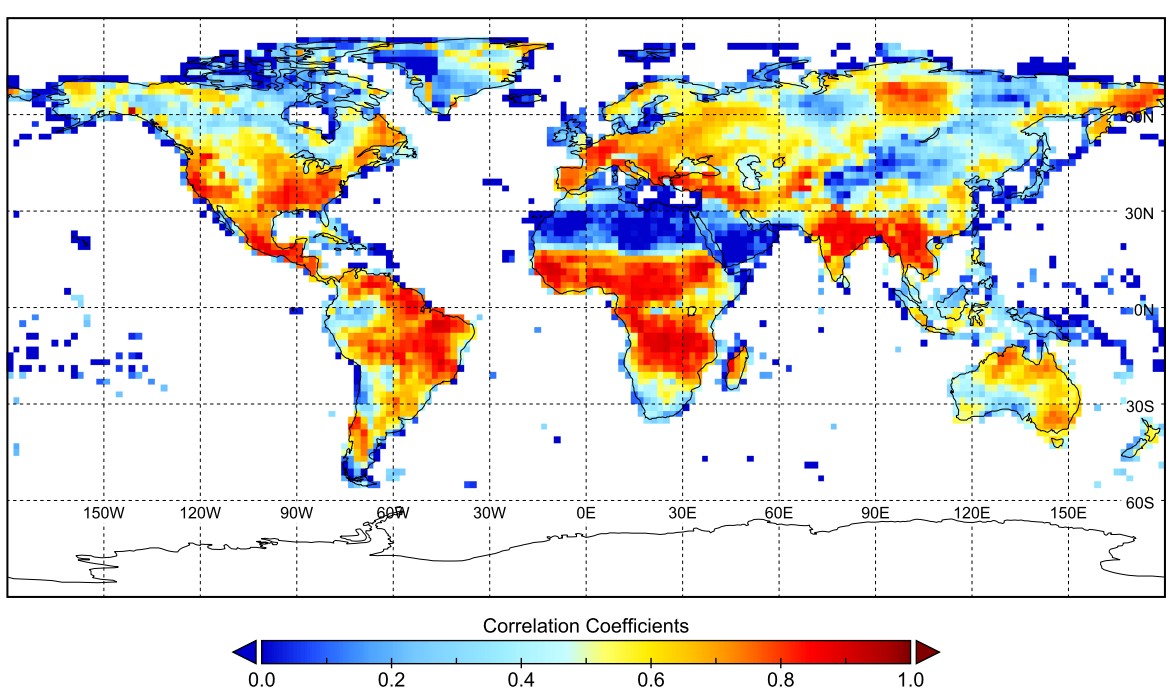

**Figure 5.** Correlation coefficients between time series of terrestrial water storage changes simulated by CoLM and those observed by GRACE satellite.

Figure 7 illustrates the global distribution of root mean square errors (RMSEs) in the time series of TWS changes simulated by CoLM compared to those observed by the GRACE satellite. Without parameter optimization, the simulated TWS changes display relatively large RMSEs compared to the observations, particularly in the Amazon basin and certain coastal areas (such as the northwestern coast of North America, surrounding regions of Greenland, and southern areas of Asia).

Difference in Correlation Coefficients (With DA minus Without DA)

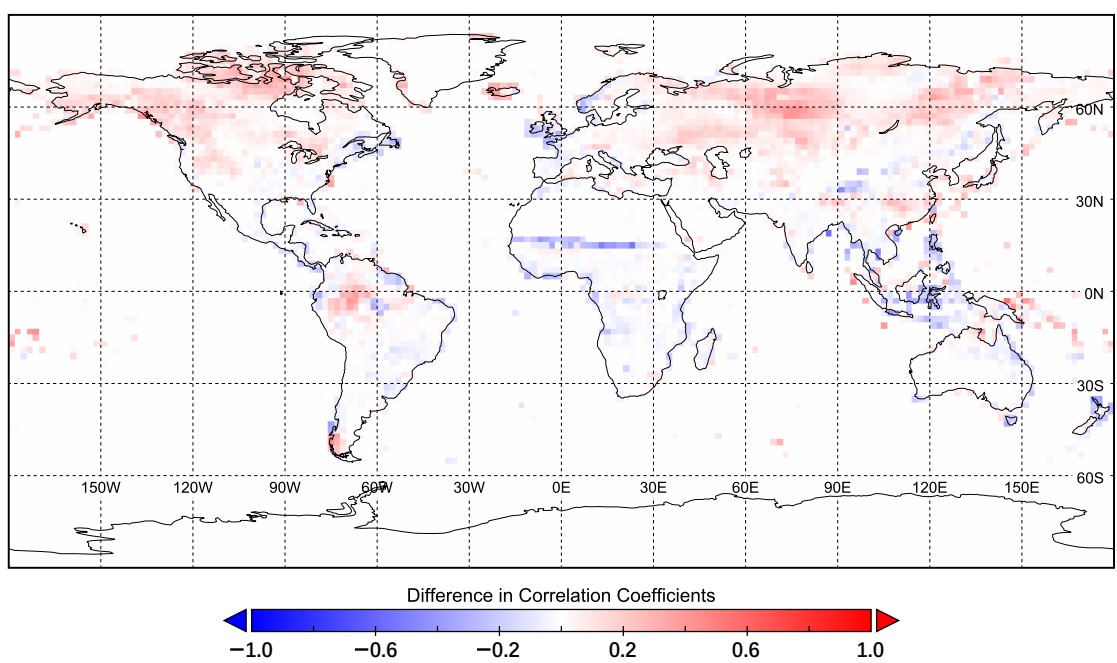

**Figure 6.** Difference between correlation coefficients by assimilating GRACE data into CoLM and those without data assimilation.

Root Mean Square Error - Without Data Assimilation

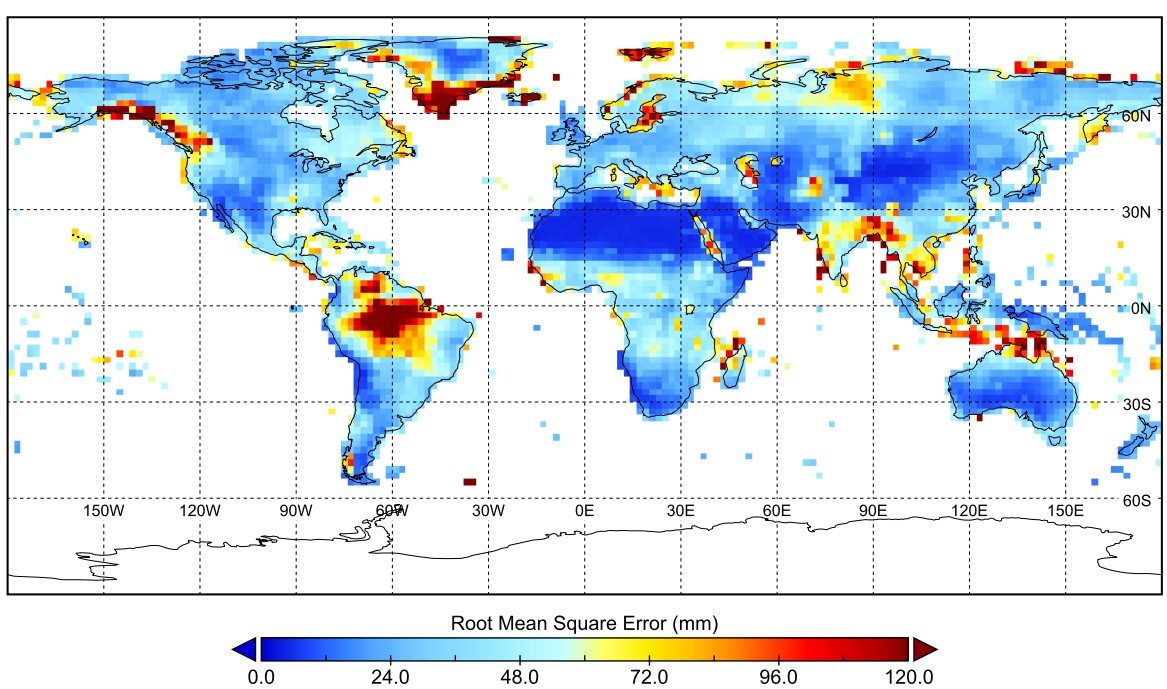

**Figure 7.** Root mean square errors in time series of terrestrial water storage changes simulated by CoLM relative to those observed by GRACE satellite.

Figure 8 depicts the differences in RMSEs resulting from assimilating GRACE data into CoLM compared to simulations without data assimilation. Through parameter optimization, approximately 65% of grid points globally experience a reduction in RMSEs, while around 33% exhibit an increase in RMSE. Specifically, significant decreases in RMSE are

observed in most parts of the Amazon basin and areas near 60° N. Conversely, noticeable increases in RMSE are noted on islands near the equatorial Pacific.

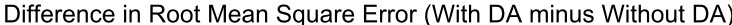

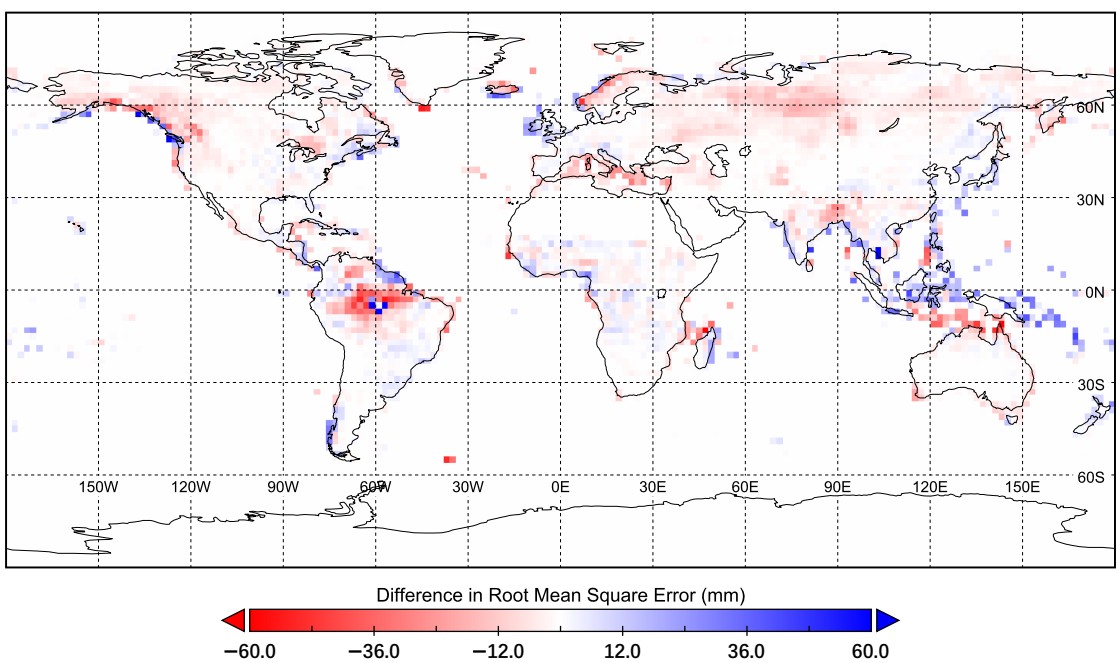

**Figure 8.** Difference in root mean square errors by assimilating GRACE data into CoLM compared to those without data assimilation.

Globally, except for arid regions, simulations of TWS changes by CoLM show good agreement with GRACE satellite observations. In most regions at mid to low latitudes, the correlation coefficients of the time series exceed 0.5, reflecting the model's performance in simulating hydrological processes in these areas. However, in high latitudes near 60° N, the correlation coefficients are lower, indicating the need for improvement in the model's simulation of hydrological processes in these regions. By using GRACE for parameter optimization, the model simulations improved in over 60% of the global regions (with increased correlation coefficients or decreased RMSE), especially in high-latitude regions near 60° N.

In arid regions, the correlation coefficient between the simulated TWS changes, and the GRACE satellite observations are generally low, but RMSE is also small. This is mainly due to the small magnitude of TWS changes in these regions, resulting in a relatively small runoff. Therefore, optimizing parameters in these areas may not be very meaningful.

A total of 13 regions globally were selected to further analyze the results of the parameter optimization, covering various climatic, topographic, and geological characteristics. The locations and extents of these regions are depicted in Figure 9.

Table 1 presents the averaged correlation coefficients and RMSE for these regions. With parameter optimization, the RMSE decreased in 12 regions except for Region 2 (the Indochinese Peninsula). Among the 13 regions, correlation coefficients improved with parameter optimization in 8 regions, while they decreased in 5 other regions. The optimization results in some regions significantly corrected biases in extreme values (not shown here).

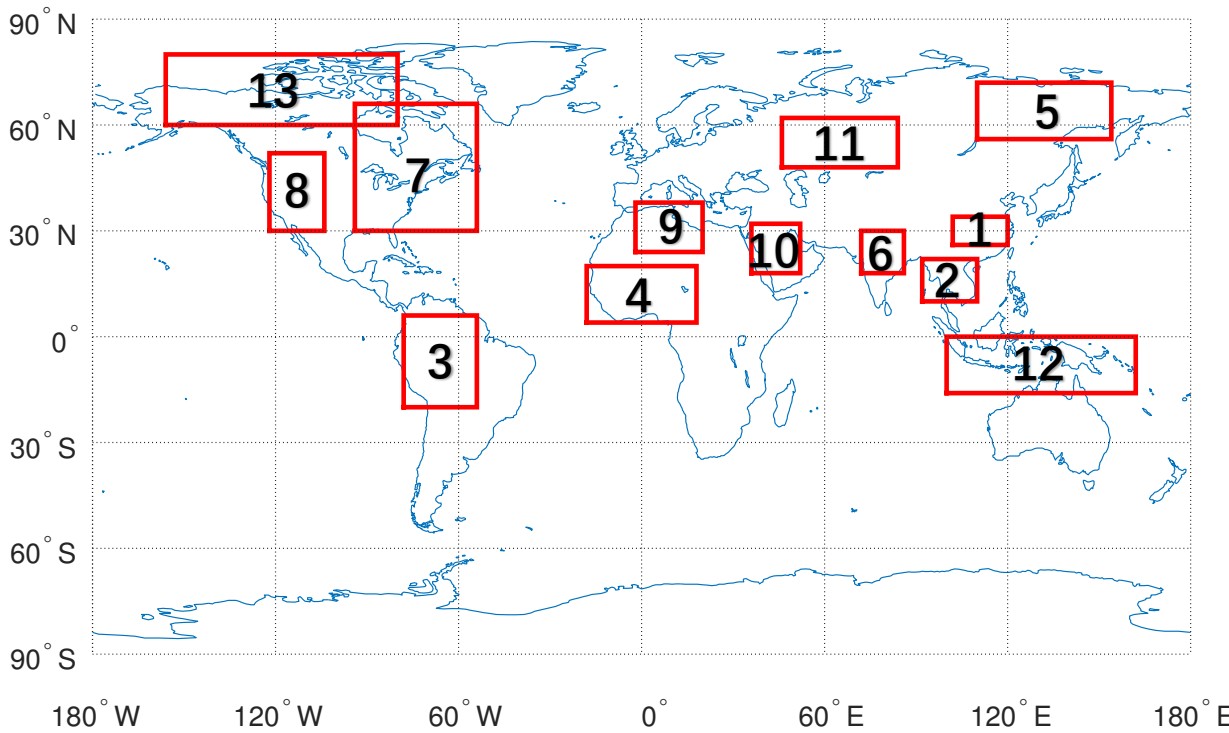

**Figure 9.** A total of 13 regions selected for further evaluations.

**Table 1.** Parameter optimization results in 13 selected regions.

| Number | Description | r without DA * | r with DA * | RMSE without DA + | RMSE with DA + |
|:---:|:---:|:---:|:---:|:---:|:---:|
| 1 | Middle and Lower Reaches of the Yangtze River | 0.478 | 0.668 | 29.2 | 24.7 |
| 2 | Indochina Peninsula | 0.930 | 0.856 | 27.8 | 40.3 |
| 3 | Northern South America | 0.943 | 0.902 | 45.1 | 30.5 |
| 4 | Central Africa | 0.957 | 0.963 | 13.2 | 11.0 |
| 5 | Northeast Asia | 0.463 | 0.585 | 25.2 | 22.0 |
| 6 | Tibet and Indian Peninsula | 0.954 | 0.950 | 34.7 | 27.6 |
| 7 | Eastern North America | 0.756 | 0.822 | 20.3 | 17.1 |
| 8 | Western North America | 0.882 | 0.920 | 15.8 | 14.3 |
| 9 | Northern Africa | 0.132 | 0.129 | 11.2 | 9.3 |
| 10 | Middle East | 0.360 | 0.367 | 11.7 | 11.7 |
| 11 | Central Eurasia | 0.584 | 0.786 | 26.3 | 18.5 |
| 12 | Islands Near the Equatorial Pacific | 0.667 | 0.647 | 24.9 | 16.5 |
| 13 | Northern North America | 0.537 | 0.708 | 19.3 | 16.0 |

* r refers to correlation coefficient. + RMSE refers to root mean square error. DA refers to data assimilation. Numbers with underline indicate larger r or smaller RMSE.

Figure 10 shows details of parameter optimization results in Region 8 (western North America). The model simulations agree well with the monthly variations observed by GRACE (Figure 10a). With data assimilation, there is a further improvement in the correlation coefficient between the model-simulated TWS changes and observations. During winter months, runoff decreases by data assimilation, while it increases during summer months (Figure 10b). With assimilation, groundwater levels decline year by year (Figure 10b). $\log \alpha_{\mathrm{slp}}$ exhibits both positive and negative values, with correlation coefficients showing improvement in most regions and root mean square errors decreasing (Figure 10c–e).

Since the changes in TWS obtained from the analysis in the optimization algorithm are not used to update the model state of TWS but only to calculate adjustment factors, the improvement in forecast accuracy comes from adjustments to runoff.

The river discharge data from GRDC serve as independent observations to assess the optimization results. The Nash–Sutcliffe efficiency (NSE) coefficient, defined in Equation (14), is employed for evaluation. The difference in NSE coefficients between simulations with parameter optimizations and those without optimizations is illustrated in Figure 11. It can be observed that for most sites in western North America, particularly around the Rocky Mountains, the NSE coefficients increase with optimization. However, in eastern North America, Europe, and the Amazon River catchment, the NSE coefficients decrease.

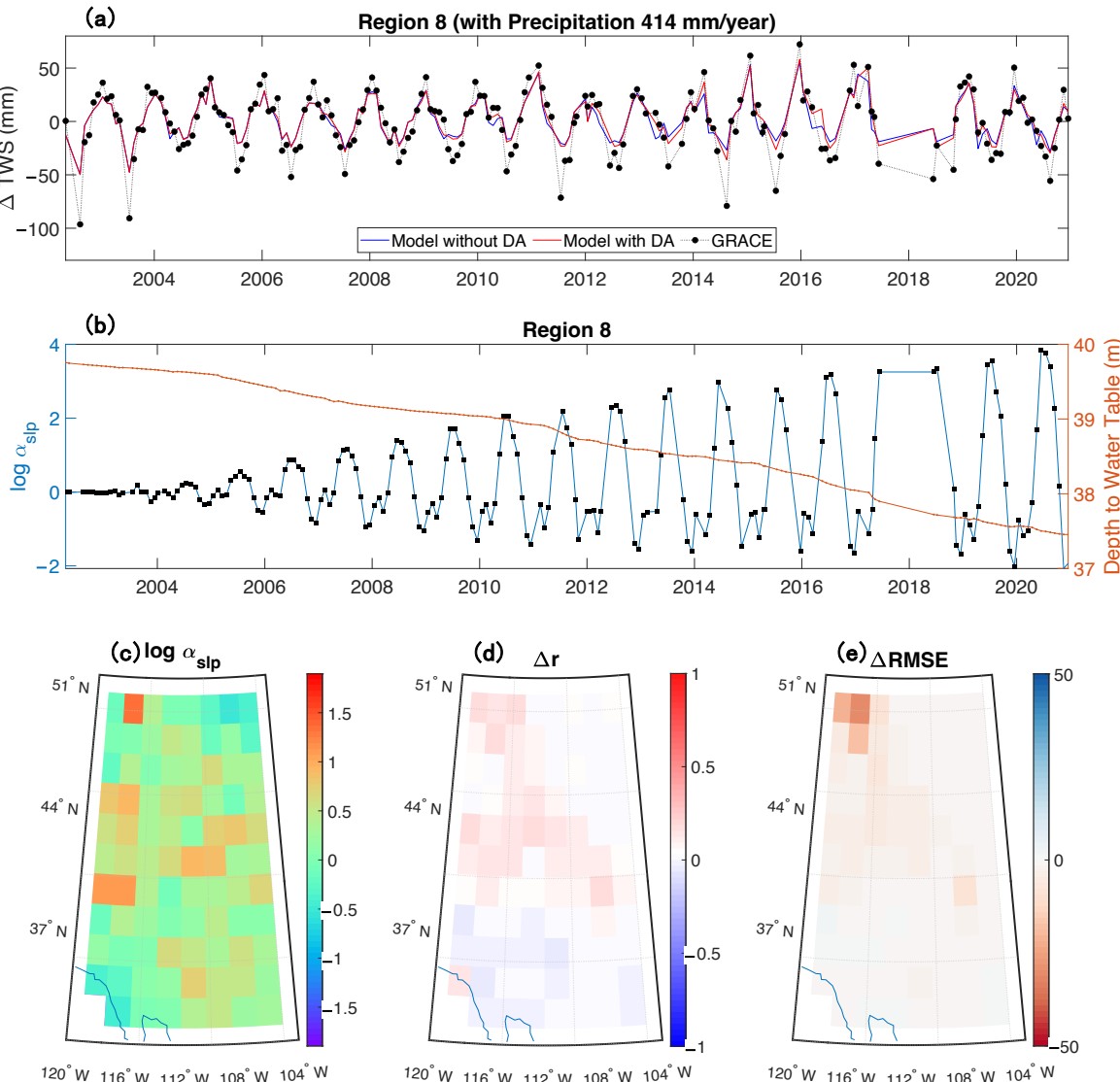

**Figure 10.** Parameter optimization results in western North America. (**a**) Time series of terrestrial water storage changes by CoLM without data assimilation, CoLM with data assimilation, and GRACE observations. (**b**) Time series of logarithm of scaling factor and change in water table depth. (**c**) Spatial distribution of time-averaged logarithm of scaling factor. (**d**) Difference in correlation coefficients between CoLM simulations with data assimilation and those without data assimilation. (**e**) Difference in root mean square errors between CoLM simulations with data assimilation and those without data assimilation.

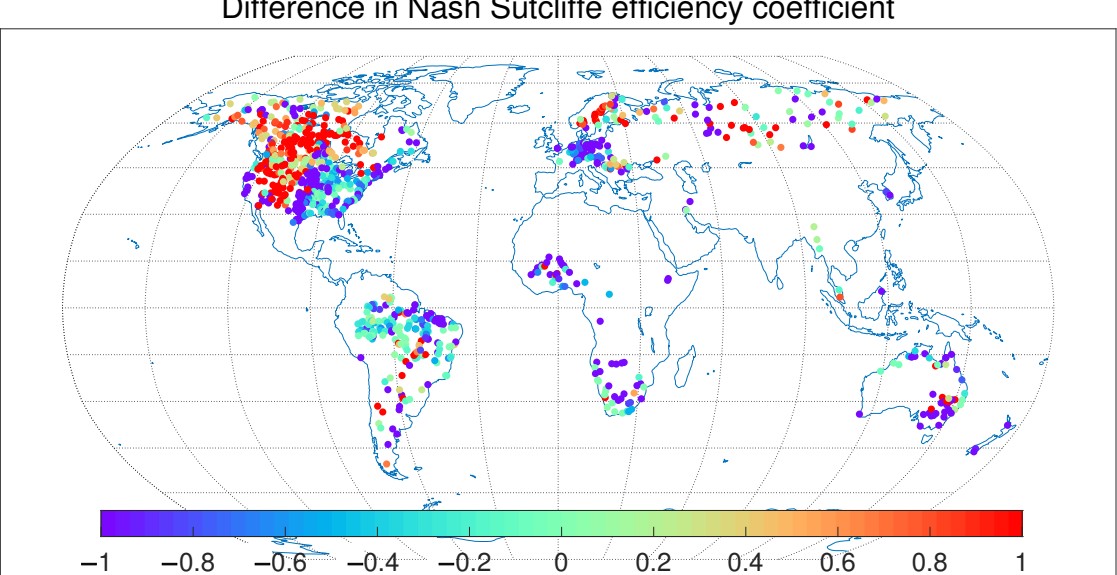

**Figure 11.** Difference in Nash–Sutcliffe efficiency coefficient between coupled model with data assimilation and that without data assimilation.

## 4. Conclusions

Land surface models play an important role in the research on terrestrial water storage. They have a broad coverage and are suitable for various scales, allowing for simulations over long periods, both past and future. However, different models employ distinct parameterization schemes, leading to uncertainties in simulation results. Addressing these uncertainties is a critical aspect of model development. Data assimilation and parameter optimization are important methods for reducing model uncertainties. Satellite observations provide essential data for optimizing large-scale global models.

In this study, within the framework of data assimilation algorithms, GRACE satellite data were applied to optimize parameters of the Common Land Model, adjusting the runoff parameterization scheme while maintaining water balance constraints. Firstly, a method for calculating the optimal value of terrestrial water storage changes within a probabilistic framework was developed, combining with maximum likelihood estimation to assess model errors. Secondly, the runoff parameterization scheme was adjusted using the optimal value of terrestrial water storage changes. This method assumes that the error in terrestrial water storage changes mainly stems from runoff. By comparing forecasted and optimal values of terrestrial water storage changes, temporally and spatially varying adjustment factors for runoff can be derived and updated online during model runs. These factors can be regarded as "empirical knowledge" learned by the model from observations.

The evaluation results indicated that after optimizing the runoff parameterization scheme using GRACE satellite observations, the correlation coefficient between the simulated and observed terrestrial water storage change improved in over 60% of global grid points. Furthermore, the root mean square error between model predictions and GRACE observations decreased in 11 out of 13 selected typical regions, with significant corrections to extreme forecast biases observed in some areas. In summary, the study demonstrates that utilizing GRACE satellite observations for parameter optimization in land surface models is effective, providing an improved simulation accuracy and reducing uncertainties in modeling terrestrial water storage changes. Independent assessments utilizing in situ river discharge observations showed notable improvements in rugged terrains like western North America.

The adjustment factors reflect some factors not considered in the runoff calculation of the model, which may vary over time and space and may be related to factors such

as topography, geology, and climate. A comprehensive examination and understanding of these factors are essential. Therefore, it is necessary to introduce other independent observations for further validation.

**Author Contributions:** Conceptualization, S.Z. and Y.S.; methodology, S.Z.; validation, Y.S.; formal analysis, Y.S.; writing—original draft preparation, Y.S.; writing—review and editing, S.Z.; visualization, Y.S.; supervision, S.Z. All authors have read and agreed to the published version of the manuscript.

**Funding:** Project supported by Southern Marine Science and Engineering Guangdong Laboratory (Zhuhai) (No. SML2023SP216). This research was funded by the Guangdong Major Project of Basic and Applied Basic Research (2021B0301030007), the Natural Science Foundation of China (under Grants 42075159).

**Data Availability Statement:** The raw data supporting the conclusions of this article will be made available by the authors on request.

**Acknowledgments:** GRACE/GRACE-FO Mascon data are available at https://grace.jpl.nasa.gov (accessed on 3 April 2024). CRU JRA forcing data are from Centre for Environmental Data Analysis. River discharge data are available at Global Runoff Data Centre website https://portal.grdc.bafg.de/ (accessed on 3 April 2024).

**Conflicts of Interest:** The authors declare no conflicts of interest.

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
