# Peer review of "Optimizing Parameters in the Common Land Model by Using Gravity Recovery and Climate Experiment Satellite Observations"

_land, doi:10.3390/land13040508_

Round 1
Reviewer 1 Report
Comments and Suggestions for Authors
Dear Authors,
I reviewed your manuscript and I found the topic interesting and easy to read. However, I would like to examine the following for the manuscript to be ready for publication.
1 - In section 2.1, the authors should mention the actual time of the experiment and how it is consistent with the GRACE for the purpose of the data assimilation.
2 - In section 2.1, why the authors run the CoLM with 2 degrees, was there a possibility to run the CoLM with 0.5 or 0.25 degrees since the terrestrial hydrological processes are sensitive to the horizontal grid spacing (such as the GLDAS in 0.25 degrees or the NLDAS in 0.125 degrees)?
3 - Information about the horizontal grid spacing of the GRACE as well as brief information about the GRACE must be given in section (2.2) especially the time availability of the GRACE.
4 - Lines 166-168 must be moved to section 2.1.
5 - A brief information about GRDC dataset must be given in section 2.2.
6 - The authors should analyze the results of figures 2 and 3 in some details.
7 - In lines 224-231, can the poor performance (noted in some regions across the globe) be attributed to the low grid spacing of the CoLM simulation (2 degrees)? Is there a possibility to run the CoLM with high-resolution (such as 0.5 or 0.25 degrees) and re-evaluate the results? Or TWS is insensitive to choice of the grid spacing? Please check this point.
You made a good job. Good luck with the revised version.
Best Regards
Author Response
We would like to sincerely appreciate the reviewer and his valuable comments. We have revised the manuscript according to these comments and provide replies to each point in the attached file.

Reviewer 2 Report
Comments and Suggestions for Authors
The manuscript presents an interesting research of integrating GRACE-derived terrestrial water storage change and the CoLM simulation outputs to improve the runoff estimation. However, I am confused about the term "data assimilation" and "parameter optimization.
Major concerns
In my opinion, the study has no data assimilation as the analysis has not been used to update the model state. The study is also not a parameter optimization study since the parameters in the runoff parameterization have not been changed. It is better to reframe the study and adjust the title accordinately.
A diagram showing the workflow would help the readers to understand the methodology.
Comments on the Quality of English LanguageAn revision would improve the readability.
Author Response

(The authors gave the same response as above.)

Reviewer 3 Report
Comments and Suggestions for Authors
MAIN REVIEW:
Basically, this study GRACE satellite data are applied to improve the parameterization of the Common Land Model, adjusting the runoff parameterization scheme with constrains on the water balance. The basic concept is good, and contributes effectively to the field of hydrological modelling.
However, there are many arbitrary steps in the methodology, which are not really analyzed or explained in details, and many of them seems to be arguable. For example, when they combine maximum likelihood estimation to assess model errors, the independency of the models is assumed, and the expectation value of their difference is assumed to be 0. Why? There is no ground on defining such assumptions. If you have any actual reason on that, you should clarify them in the paper. Furthermore, they have decided to change purely the runoff parameters to set the optimal value of terrestrial water storage changes. Again, I cannot see any reason to assume inferiority of the runoff (or superiority of the evapotranspiration) in the accuracy of their knowledge. Why not to re-scale the evapotranspiration?
Additionally, the choice of some models, and their inconsistency is arguable. For example, in one of the official documentations of the CRU JRA reanalysis product (which the authors have used) is https://dx.doi.org/10.2151/jmsj.2015-001 (Kobayashi et al, 2015), they state: "(...) great caution is needed when using hydrological variables from reanalyses, especially model diagnostics such as precipitation and evaporation". I do agree with this statement. Still, you can use a reanalysis product, for sure. However, a reanalysis model is conserved for the energy. Consequently, all variables are tied to each other, they cannot be considered to be independent. So if you take the precipitation from a reanalysis product, you should take the other components of the hydrological cycle from the same source. In line 68-69 you say: "Precipitation is obtained from atmospheric forcing data, while evapotranspiration and runoff are simulated by the model." So precipitation is from CRU JRA, but not for evapotranspiration and runoff are from CoLM, right? This is an arguable choice.
Finally, the results are not fully convincing. Some of the validation tests does not seem to be accurate. Yes, they show some improvements, which is nice, but additional improvements could have been achieved by refining the methodology. Although the topic of the study is relevant, the aim of the study to integrate models of different sources is very important and ambitious, I am not convinced in the correctness of the fundamentals and procedure of the modelling methodology.
MINOR COMMENTS:
Lines 19-26: The use of the word "flux" is wrong. I am not sure, but probably You are referring to the different components of the hydrological cycle, the continuous circulation of water mass in the geosphere-atmosphere system. You also might refer to geophysical fluids, so fluids in wider sense, and particularly to water. I am not native English, so I don't know, what would be the correct term. Flow? Stream? Fluid? Anyway, flux is not good, definitely.
Lines 27-28: "In land surface models, there are primarily two approaches for simulating runoff: parameterization schemes and explicit incorporation of lateral flow dynamics." Well, I am not sure about it. Please, support your statement with a reference!
Lines 29-31: "However, this method necessitates data and parameterization schemes at the hillslope scale [6,10], which are currently limited [11]." Actually, the referred study [11] is about hyperresolution, which is appr. 1 km or below. I would not consider hillslope scale to belong this scale, it can be on the 1-10 km range.
Line 62: "Common Land Model (CoLM)" This abbreviation has already been introduced in the Introduction paragraph, so no need to introduce it again.
Equation (2)-(5): Did you defined these equations particularly for this study? It doesn't seem so. So please, provide a reference publication, where the rationale behind the functions and the used parameterization is described. Or, provide this information in this paper.
Line 85-86: "Gwat represents the amount of liquid water reaching the surface, including rain through the canopy, water dripping from canopy, and snowmelt" It is an integration of too complex processes... how can you estimate or model it?
Line 100: First time you use an abbreviation, please provide the full name. Actually, beyond the lack of the full name, I also miss the description of the CRU JRA forcing data. Apart from the information in the Acknowledgments, that this data is from Centre for Environmental Data Analysis, no additional information is provided. There is no link to the data, no information on the used version number (is it the V2.3 dataset?), exact definition of the used variable, a proper citation of the dataset, (see https://doi.org/10.1038/s41597-020-0453-3 or https://dx.doi.org/10.2151/jmsj.2015-001 , depending on the used version of the data).
Line 106-108: "The near-equilibrium state is determined based on the criterion that the ten-year moving average of the change in ΔW is less than 2% of the annual precipitation." How the threshold of 2% has been chosen? And why is it tied to precipitation, why not the total water storage?
Line 112: "GRACE-FO MASON RL06Mv2"... correctly it is "GRACE-FO MASCON RL06Mv2"
Line 120-121: "with uncertainties of σo0 and σo1, respectively"... Where these uncertainty information is taken from? If it is purely the model error provided to the MASCON solution, then it might not include all relevant error sources.
Equation (8), right-hand side: In order to fix mass conservation, a scale factor is applied on runoff. What is the rationale behind applying this scale factor on the runoff and not on the evapotranspiration? From in-situ measurements surface runoff known quite precisely, but evapotranspiration and subsurface runoff not that easily measurable. Why you were choosing purely the runoff and neglecting evapotranspiration?
Equation (8), left-hand side: You determine the ΔWa/Δt ratio. In the case of the GRACE and GRACE-FO models, how did you determined the interval of Δt? Is it the last day minus first day? OR Is it the number of days have been used to determine that certain model? OR simply all models were considered to refer to 30 or 31 day long periods? All of the approaches are arguable... however, whatever you did, this issue should be discussed at least.
Line 136-141, the online updating scheme: this is not clear for me. In equation (8), ΔWa refers to the change between two consecutive months, m and m+1. According to equation (8), there is only one value of αslp is referring to two epochs. So why do you write here about two adjustment factors αslp referring to epochs m and m+1? Clarify the description of this step!
Line 143: "Assuming ΔWf − ΔWo is a random variable with mean 0 and variance (σf)^2 + (σo)^2"... The ΔWf − ΔWo is not a random variable, these are two models including all their systematic modelling errors, etc., so why should be mean of their difference equal to 0? Also, their variance in your assumption is that these models are independent. This can also be argued, but for me it is OK, you have no better assumption at the beginning.
Line 166-167: "To evaluate data assimilation results with independent observations, CoLM is coupled with CaMa-Flood [36] to simulate discharge in rivers." If I understand correctly, you couple two models (CoLM and CaMa-Flood) for determining the discharge of the rivers. When you use reanalysis products, you can get all relevant hydrological variables in a consistent system, and you don't have to couple different models with each other, involving all systematic errors (essentially enlarged by artificially restricting the summarized water budget) of such a model.
Equation (15): it is not clear for me, why do you take the temporal average of the logarithm of the αslp multiplicators, and not that of the multiplicator itself? Why is it more reasonable to choose the logarithm, then the values themselves?
Figure 1: the logarithm of the multiplicator factor varies between -4 and +4, so (in case this logarithm here is the common logarithm meaning logarithm with base 10) it refers to values in the 0.0001 to 10,000 range. This is a huge range. It means that the runoff data used for the input is simply wrong from the aspect of the modelling.
Figure 8: the choice of the 13 regions seems to be improper. GRACE and GRACE-FO mass variations can be considered to be representative, if it is properly incorporating regions with the same climatic properties. Therefore, when a river basin is analyzed, it is adequately masked and numerically isolated from the effect of the nearby mass variation processes. In the case of your 13 regions, they seem to integrate land and ocean area into one test area. For such selections I would not expect any informative result, indicating that your assimilation of GRACE models to CoLM is providing something, which is numerically an improvement, but physically may not indicate anything. For sake of clarification: I don't mean that your modelling (assimilation of GRACE data) is wrong. I just mean that this test is not correct.
Figure 10: I have limited experience with the Nash Sutcliffe efficiency coefficient. But as much as I know about it, when it's value is below for example 0.5, the results is pretty bad (since the Nash Sutcliffe efficiency coefficient describes 1 minus error variance over signal variance, so for these cases the error is in comparable range with the signal). Based on this (and sorry if I misinterpret your results) most of the test sites seems to be pretty bad (all, which have no orange of red colour). If this interpretation is wrong, would you explain in the text how to interpret the results on Figure 10 correctly?
Lines 267-269: "However, different models employ distinct parameterization schemes, leading to uncertainties in simulation results. Addressing these uncertainties is a critical aspect of model development." Yes, this is the key issue. And this is the aspect of the manuscript, which could be improved. Check modelling error aspects of https://doi.org/10.1016/j.jhydrol.2021.126202 or https://doi.org/10.3390/w15091725 , among other studies.
Line 293-297: "The adjustment factors reflect some factors not considered in the runoff calculation of the model, which may vary over time and space and be related to factors such as topography, geology, and climate. A comprehensive examination and understanding of these factors are essential. Therefore, it is necessary to introduce other independent observations for further validation." Yes, I agree, it is an essential aspect. But as you cannot assume independency of observations in a complex climate system, instead of introducing new variables into your solution considering their independence, you should rather make attempts to determine and numerate the covariance of the involved variables.
Author Response

(The authors gave the same response as above.)

Round 2
Reviewer 1 Report
Comments and Suggestions for Authors
Dear Authors,
I checked the revised version of the manuscript and I see that you made good efforts and responded to my comments properly. Therefore, I see that the manuscript is suitable for publication in its present form. Just wanted to ask you to revise Figure 1 in the revised version because it has been removed.
Congratulations.
Best Regards
Reviewer 3 Report
Comments and Suggestions for Authors
Any scientific work or research report can be improved indefinitely. The authors have responded appropriately to most of the corrections I have requested, and the requested changes have been accepted. Accordingly, I consider the article suitable for publication in its present state.